# Hot Deformation Behaviour and Constitutive Equation of Mg-9Gd-4Y-2Zn-0.5Zr Alloy

**DOI:** 10.3390/ma15051779

**Published:** 2022-02-26

**Authors:** Yangjunfeng Nie, Jie Zheng, Rui Han, Leichen Jia, Zhimin Zhang, Yong Xue

**Affiliations:** School of Materials Science and Engineering, North University of China, Taiyuan 030051, China; nyjfjsy@126.com (Y.N.); cqzhengjie@163.com (J.Z.); hanrui19980914@163.com (R.H.); jlc226688@hotmail.com (L.J.); zhangzhimin@nuc.edu.cn (Z.Z.)

**Keywords:** Mg–RE alloy, hot compression, constitutive relation, DRX

## Abstract

The thermal deformation behaviour of Mg-9Gd-4Y-2Zn-0.5Zr alloy at temperatures of 360–480 °C, strain rates of 0.001–1 s^−1^ and a maximum deformation degree of 60% was investigated in uniaxial hot compression experiments on a Gleeble 3800 thermomechanical simulator. A constitutive equation suitable for plastic deformation was constructed from the Arrhenius equation. The experimental results indicate that due to work hardening, the flow stress of the alloy rapidly reached peak stress with increased strain in the initial deformation stage and then began to decrease and stabilize, indicating that the deformation behaviour of the alloy conformed to steady-state rheological characteristics. The average deformation activation energy of this alloy was Q = 223.334 kJ·mol^−1^. Moreover, a processing map based on material dynamic modelling was established, and the law describing the influence of the machining parameters on deformation was obtained. The experimental results indicate that the effects of deformation temperature, strain rate and strain magnitude on the peak dissipation efficiency factor and instability range were highly significant. With the increase in the strain variable, the flow instability range increased gradually, but the coefficient of the peak power dissipation rate decreased gradually. The optimum deformation temperature and strain rate of this alloy during hot working were 400–480 °C and 0.001–0.01 s^−1^, respectively.

## 1. Introduction

Magnesium (Mg) alloys have become a popular lightweight structural material and are widely used in automobiles, high-speed railways and aircraft [1,2,3,4,5]. However, it is well-known that Mg alloys have poor plasticity and formability at room temperature because of their dense hexagonal close-packed structure [6,7]. There are relatively few slip planes and slip directions compared with body-centred cubic and face-centred cubic systems, so there are fewer slip systems, which limits their application. Therefore, plastic deformation at high temperature is generally applied to increase the number of slip systems and reduce the critical shear stress of the non-basal plane slip to improve plasticity and enhance workability [8].

In recent years, an increasing number of studies have found that adding rare earth (RE) elements into Mg alloys can improve the properties of the alloys at both room and high temperature [9,10,11,12,13]. After deformation, this process can form fine crystalline materials. In this study, a constitutive equation was introduced to better study the hot deformation behaviour of Mg–RE alloys at high temperature, and different deformation conditions (e.g., temperature, strain rate and deformation degree) were changed to explore the deformation mechanism of Mg–RE alloys. Sun et al. studied the microstructure, hot deformation behaviour, texture evolution and processing map of Mg-8Sn-1.5Al (wt.%) alloys [14]. Temperature of 603–633 K and strain rate of 0.03–0.005 s^−^^1^ were projected to be the best process parameters using the constitutive equation and hot processing map. Bao et al. constructed the processing map of Mg-6Zn-5Ca-3Ce (wt.%) alloy based on the Murty criterion [15]. According to the constitutive equation and hot processing map, the optimal parameters were T = 590–640 K, $\dot{\varepsilon }$ = 0.0001–0.0003 s^−^^1^ and T = 650–670 K, $\dot{\varepsilon }$ = 0.0003–0.004 s^−^^1^. Wang et al. investigated the constitutive behavior and hot working properties of Mg-4Al-2Sn-Y-Nd (wt.%) alloy at 200–400 °C with a strain rate of 1.5 × 10^−^^3^–7.5 s^−^^1^ [16]. The results show that this alloy was suitable for hot working at high strain rates and that the main deformation mechanism involved dislocation climbing at high temperatures. Yu et al. conducted hot compression experiments on a new quaternary alloy, Mg-6Zn-1.5Cu-0.5Zr (wt.%) [17]. The hot deformation behaviour of the alloy at a temperature of 523–673 K and strain rate of 0.001–1 s^−1^ was used to predict the flow stress with a constitutive equation and hot processing map. The predicted results agree well with the data measured in the preliminary experiment. The safe zone (i.e., dynamic crystallization zone) and dangerous zone (i.e., cavity zone and grain boundary crack zone) of material plastic processing can be judged by the hot working diagram. This enables realization of the reasonable development of material hot working technology and precise control of material microstructure. However, few studies on the hot working characteristics of Mg–RE alloys have been conducted in the past, especially on the constitutive behaviour and hot working properties of Mg-9Gd-4Y-2Zn-0.5Zr Mg–RE alloys under compressive deformation. Without an in-depth study on the deformation softening and instability zone at high temperature, it is impossible to formulate an effective hot processing parameter.

Therefore, in this experiment, Mg-9Gd-4Y-2Zn-0.5Zr alloy was hot compressed in a Gleeble 3800 system (manufacturer, city, country). Data were obtained through multiple experiments, and true stress–true strain curves and processing diagrams were drawn. The alloy’s hot deformation behavior and hot working properties were studied at deformation temperatures of 360 to 480 °C and strain rates of 0.001 to 1 s^−1^. The constitutive equation was established based on the data obtained in the experiment, and the optimal hot working conditions of this alloy were determined from a combination of the microstructure and processing map. This work provides real and effective data for the thermoplastic deformation and subsequent hot forming of this alloy.

## 2. Methods and Experimental Section

The as-cast bar used in this experiment was Mg-9Gd-4Y-2Zn-0.5Zr (wt.%) alloy. Figure 1 shows the original structure of the alloy and Table 1 shows the chemical composition. Figure 1a shows that the original structure of this Mg–RE alloy was composed of a matrix phase and a eutectic phase; in the figure, the matrix phase is black, and the eutectic phase is distributed along the grain boundaries of the matrix phase. The fishing network characteristic of the eutectic phase was a result of the rapid cooling of Mg–RE alloy in the casting process. RE compounds leading to high melting points were not sufficiently diffused, and a considerable amount of segregation occurred at the grain boundaries to form eutectic structures [18]. The partial enlargement of Figure 1a shows that there were some discontinuous particles in the eutectic structures, which were RE-rich phases. The as-cast microstructure of this Mg–RE alloy was coarse and irregular, with an average grain size of 70 μm.

A cylindrical ingot was obtained by cutting a section of the disk blank along the transverse direction. The disk blank was machined, and a cylindrical sample with a diameter of Φ8 mm × 12 mm was removed from the axial direction. To prevent severe barrelling of the specimen, the upper and lower sides of the sample were coated with high-temperature-resistant molybdenum disulfide grease. The specimens were placed in a Gleeble 3800 thermomechanical simulator (DSI, St. Paul, Minnesota, USA) for uniaxial hot compression tests at 360, 400, 440 and 480 °C; strain rates of 0.001, 0.01, 0.1 and 1 s^−1^; and a maximum deformation of 60%. The heating rate of the hot compression test was 10 °C/s, and preheating was carried out for 3 min before compression. The purpose of heat preservation was to ensure that the temperature of each area inside and outside the sample was uniform and consistent. After compression, water cooling was applied quickly to retain the compressed structure.

The compressed sample was cut along the longitudinal centre and then etched for 5–6 s after mechanical polishing (etchant: 2 mL distilled water +2 mL acetic acid +1 g picric acid + 14 mL ethanol). The microstructure of the samples was observed with an optical microscope (OM; DM2500M, Leica Microsystems, Wetzlar, Germany) and a scanning electron microscope (SEM; SU5000, Hitachi, Tokyo, Japan). The scanning electron microscope was a field-emission scanning electron microscope with an acceleration voltage of 20 kV. After mechanical polishing, the surface stress layer was removed by ion thinning at an angle of 3.5° and an acceleration voltage of 6.5 kV. Then, the EBSD test was carried out using the scanning electron microscope (SU-5000) (Hitachi, Tokyo, Japan). The working distance was 15 mm, the acceleration voltage was 20 kV, and the angle was 70°. EBSD data were analyzed by Channel 5 software (EDAX, Philadelphia, PA, USA), including OSC file noise cleaning and IPF graph generation.

## 3. Results and Discussion

### 3.1. True Stress-True Strain Curve

Metals and alloys are strongly affected by hot activation processes during plastic deformation at high temperatures, as the flow stress is greatly affected by the temperature and strain rate. Figure 2 shows the true stress–true strain curves of the Mg-9Gd-4Y-2Zn-0.5Zr alloy at different temperatures and strain rates. Each curve is divided into three stages: the work hardening stage, dynamic recrystallization (DRX) initiation stage and stable rheological stage. The first is the work hardening stage, when true stress increased sharply with a slow increase in true strain. Then, the DRX stage began when the stress reached a critical value. With increasing deformation, the dislocation density increased, the growth rate of DRX accelerated, the softening effect strengthened, and the increase in stress slowed. When the stress reached maximum value, the softening effect of DRX exceeded the effect of work hardening, and then the stress decreased with increasing strain. Finally, in the stable rheology stage, with the increase in true strain, the effects of work hardening and DRX tended to stabilize, and the rheological stress tended to remain constant.

Additionally, when the temperature was constant, the strain corresponding to peak stress increased with increasing strain rate. However, when the strain rate was constant, the strain corresponding to peak stress decreased with increasing temperature. Reducing the deformation temperature and increasing the strain rate increased the slope of the true stress–true strain curve and flow stress values because these stresses could not fully disperse during the plastic deformation of the metal, and the deformation range was short, resulting in a higher critical shear stress experienced by the metal crystal; this suggests that increasing the strain rate or reducing the deformation temperature significantly impacts the strain-hardening effect. Increasing the deformation temperature and decreasing the strain rate increased the deformation-softening effect. The increase in deformation temperature provided more energy and higher grain boundary mobility, leading to the nucleation and growth of dynamically recrystallized grains and reducing the flow stress level. As the strain rate decreased, the dislocations fully spread to achieve reorganization, thus increasing the DRX degree and softening effect [19].

### 3.2. The Establishment of a Constitutive Equation

The Arrhenius equation was used in this experiment to accurately describe the relationship between flow stress and deformation parameters such as strain rate and temperature, providing a method for studying the effects of various deformation factors on the flow stress of this Mg–RE alloy during high-temperature deformation [20,21]. Equation (1) shows that the relationship between the strain rate and flow stress is a power function at low stress states. Equation (2) shows that the relationship between the strain rate and flow stress is an exponential function at high stress states. Equation (3) is a more comprehensive constitutive model proposed by Sellars et al. [22] and can be applied under any stress condition.
(1)$$\dot{\varepsilon }=A_{1}\sigma^{n1}exp\left(-Q/RT\right)\;\alpha \sigma <0.8$$
(2)$$\dot{\varepsilon }=A_{2}exp\left(\beta \sigma \right)exp\left(-\frac{Q}{RT}\right)\;\alpha \sigma >1.2\;$$
(3)$$\dot{\varepsilon }=A{\left[sinh\left(\alpha \sigma \right)\right]}^{n}exp\left(-Q/RT\right)\;for\;all\;\alpha \sigma \;$$

Here, *Q* is the deformation activation energy (J·mol^−^^1^); *T* is the deformation temperature (K); *R* is the gas constant (J·(mol·K)^−^^1^), *R* = 8.314 J·(mol·K)^−^^1^; $\dot{\varepsilon }$ is the strain rate (s^−^^1^); σ is the flow stress (MPa); *A*_1_, *A*_2_, *A*, α and β are material constants; n_1_ and n are work hardening indices and $\alpha =\beta /n_{1}$. 

In this model, the peak stress $\sigma_{p}$ is used to calculate the material parameters β, α, *n*_1_, *n*, *Q* and *A*. First, it is assumed that the deformation temperature *T* has little effect on the deformation activation energy *Q*, and *Q* is independent of *T*. The natural logarithm of both sides of Equations (1) and (2) can be obtained:(4)$$ln\dot{\varepsilon }=lnA_{1}+n_{1}ln\sigma -\frac{Q}{RT}$$
(5)$$ln\dot{\varepsilon }=lnA_{2}+\beta \sigma -\frac{Q}{RT}$$

Figure 3a,b shows the relationship between $\sigma -ln\dot{\varepsilon }$ and $ln\sigma -ln\dot{\varepsilon }$ at peak stress, and the average of the linear fitting slopes of $\sigma -ln\dot{\varepsilon }$ and $ln\sigma -ln\dot{\varepsilon }$ were calculated. Therefore, *n*_1_ and β were obtained from the fitting curve and were 7.78028 and 0.07445, respectively. Additionally, α was obtained from $\alpha =\beta /n_{1}$ and was 0.00957.

The natural logarithm of both sides of Equation (3) can be obtained:(6)$$ln\dot{\varepsilon }=lnA-\frac{Q}{RT}+nln\left[sinh\left(\alpha \sigma \right)\right]$$

Zener and Hollomon [23] introduced temperature parameters and used the *Z* parameter to represent the relationship between the deformation temperature and strain rate:(7)$$Z=\dot{\varepsilon }exp\left(\frac{Q}{RT}\right)=A{\left[sinh\left(\alpha \sigma \right)\right]}^{n}$$

When the strain rate $\dot{\varepsilon }$ is constant, Equation (7) can be differentiated at different temperatures or strain rates to obtain the expression *Q*, and the formula can be arranged to obtain Equation (8). Figure 3c,d corresponds to the $ln\left[sinh\left(\alpha \sigma \right)\;\right]-ln\dot{\varepsilon }$ and ln [sinh(ασ) ]−1/T diagrams and linear fitting.
(8)$$Q=R{\left\{\frac{\partial ln\dot{\varepsilon }}{\partial ln\left[sinh\left(\alpha \sigma \right)\right]}\right\}}_{T}{\left\{\frac{\partial ln\left[sinh\left(\alpha \sigma \right)\right]}{\partial \left(1/T\right)}\right\}}_{\dot{\varepsilon }}$$

The slope of the ln[sinh(ασ) ]−1/T curve at different strain rates was averaged, and the average value of the hot activation energy *Q* was obtained with Equation (8) as *Q* = 223.334 kJ·mol^−^^1^, which is higher than that of self-diffusion in pure Mg. Bao et al. studied the hot deformation behaviour of Mg-6Zn-5Ca-3Ce (wt.%) alloy and believed that the change in the *Q* value was due to the exceptionally fine grain size, since these fine grains could act as barriers to effectively restrict dislocation motion [15].

Temperature and deformation rate have important influences on metal deformation behaviour, and the parameter Z can characterize this influence. Logarithms of both sides of Equation (7) were taken to obtain a simplified expression:(9)lnz=lnA+nln[sinh(ασ)]

Figure 4 shows the relationship between lnZ and ln[sinh(ασ)], and *lnA* and *n* were determined to be 34.00007 and 5.00753, respectively.

The expression of Z with the temperature compensation factor can be calculated as follows:(10)$$\sigma =\frac{1}{\alpha }ln\left\{{\left(\frac{Z}{A}\right)}^{\frac{1}{n}}+{\left[{\left(\frac{Z}{A}\right)}^{\frac{2}{n}}+1\right]}^{\frac{1}{2}}\right\}$$

The constitutive equation of this alloy was obtained by substituting different material parameters into Equation (3):(11)$$\dot{\varepsilon }exp\left(\frac{223334}{RT}\right)=5.83\times 10^{14}{\left[sinh\left(0.00957\sigma \right)\right]}^{5.01}$$

However, the constitutive equation established here ignores the effect of strain, so it has some limitations. In order to more accurately reflect the effect of strain on flow stress at various strain rates, a constitutive equation under full stress should be established. The curve shown in Figure 5 used the same method to calculate the relationship between material constants (*n*, α, *Q*, *A*) and strain under different stresses. Therefore, the values of the material constants α, Q, lnA and n corresponding to different strain variables were calculated by the polynomial fitting method within the strain range of 0.1–0.9. It was found that the quintic polynomial had a strong fitting effect, and the fitting formulas were as follows:
(12)$$\alpha_{\varepsilon }=-0.04808\varepsilon^{5}+0.15574\varepsilon^{4}-0.20211\varepsilon^{3}+0.1232\varepsilon^{2}-0.02775\varepsilon +0.01222\;$$



$$n_{\varepsilon }=-21.15385\varepsilon^{5}+81.52681\varepsilon^{4}-113.52273\varepsilon^{3}+73.72203\varepsilon^{2}-23.49226\varepsilon +7.645\;$$


$$Q_{\varepsilon }=-30666.98718\varepsilon^{5}+16136.29079\varepsilon^{4}+61275.794\varepsilon^{3}-59843.06454\varepsilon^{2}\;+13822.3636\varepsilon +224105.14417\;$$


$$lnA_{\varepsilon }=33.78365\varepsilon^{5}-111.68539\varepsilon^{4}+138.10725\varepsilon^{3}-76.67347\varepsilon^{2}+18.49031\varepsilon +\;32.47914.$$



Substituting material constants $\alpha_{\varepsilon }$, $n_{\varepsilon }$, $Q_{\varepsilon }$ and $lnA_{\varepsilon }$ into Equation (3), the Arrhenius constitutive equation considering strain can be obtained:(13)$$\dot{\varepsilon }=A_{\varepsilon }{\left[sinh\left(\alpha_{\varepsilon }\sigma \right)\right]}^{n_{\varepsilon }}exp\left(-Q_{\varepsilon }/RT\right)$$

### 3.3. Error Analysis

The accuracy of the Arrhenius constitutive equation established above was tested by introducing correlation coefficient (*R*) and absolute average relative error (*AARE*) formulas corresponding to the calculated stress and experimental stress under the strain condition of 0.7.
(14)$$R=\frac{\sum_{i=1}^{N}\left(X_{i}-\bar{X}\right)\left(Y_{i}-\bar{Y}\right)}{\sqrt{\sum_{i=1}^{N}{\left(X_{i}-\bar{X}\right)}^{2}}\sqrt{\sum_{i=1}^{N}{\left(Y_{i}-\bar{Y}\right)}^{2}}}$$
(15)$$AARE=\frac{1}{N}\overset{N}{\underset{i=1}{\sum }}\left|\frac{Y_{i}-X_{i}}{X_{i}}\right|\times 100\%$$
$\bar{X}$ and $\bar{Y}$ are the averages of the experimental and calculated values, respectively; the number of experimental data points studied is represented by *N*.

The correlation coefficient *R* is a quantity used to evaluate the degree of linear correlation between experimental values and calculated values, and the *AARE* is a nonbiased statistical parameter used to judge the precision of the constitutive equation. In the formula above, $X_{i}$ and $Y_{i}$ are the experimental value and the calculated value of stress under the strain condition of 0.7, respectively.

A stress scatter diagram was established with the experimental value as the abscissa and the calculated value as the ordinate, as shown in Figure 6. The diagonal line is the ideal 45° diagonal. The correlation coefficient between the experimental value and the calculated value, *R* = 0.96696, was calculated with Origin data processing, indicating a good correlation between the calculated value and the experimental value under the strain condition of 0.7. The calculated AARE is 17.915%, which shows that the established constitutive equation could better predict the stress of the alloy during hot deformation.

### 3.4. Processing Map

Based on dynamic material model (DMM) theory, Nayan et al. established a hot working diagram composed of an energy dissipation diagram and an instability diagram. The hot working diagram has played a very important role in analysing hot deformation behaviour and optimizing hot working processes [24]. The hot working diagram records the plastic deformation capacity of materials under different hot deformation conditions (deformation temperature, strain rate and strain variables). The safe zone (i.e., the dynamic crystallization zone) and dangerous zone (i.e., the cavity zone and grain boundary crack zone) of plastic processing can be determined using a hot working diagram to rationally design hot working technology for materials and accurately control the microstructure of materials. The strain-rate-sensitivity index *m* is associated with lnσ and $ln\dot{\varepsilon }$ at the corresponding temperature:(16)$$m=\frac{\partial ln\sigma }{\partial ln\dot{\varepsilon }}$$

The power dissipation rate is related to the evolution of the microstructure under different deformation temperatures and strain rates (e.g., dynamic recovery and dynamic crystallization). Each region corresponds to a specific microstructure characteristic, and a high-power dissipation efficiency indicated that this region had dynamic crystallization or local flow instability [25]. The dissipation factor η can be calculated from the strain rate sensitivity index *m*:(17)$$\eta =2\left(1-\frac{1}{1+m}\right)=\frac{2m}{1+m}$$

In a dynamic material model, the principle of machining instability was established based on the principle of the maximum entropy generation rate. According to the maximum principle of irreversible thermodynamics, the instability criterion ξ can be obtained:(18)$$\xi =\frac{\partial \ln\left(\frac{m}{m+1}\right)}{\partial ln\dot{\varepsilon }}+m<0$$

The instability parameter ξ is derived as a function of the deformation temperature and strain rate. Different flow instability diagrams can be obtained by changing the deformation temperature or strain rate, and the optimal microstructure and performance can be obtained by superposing the power dissipation diagrams at strain rates of 0.001–1 s^−1^ and deformation temperatures of 360–480 °C. Therefore, the processing map reflects the effect of hot processing parameters on the dissipation efficiency factor and instability state, which helps to avert instability regions and select the optimal hot processing parameters.

Figure 2 shows that the alloy was affected by DRX during hot deformation, and the flow stress first increased rapidly to a peak value and then decreased slowly to reach the equilibrium state. Therefore, hot working diagrams corresponding to strains of 0.3, 0.5 and 0.7 were obtained. Figure 7 shows that the contour line is the dissipation factor η. As the temperature and strain rate increased, the peak value of the dissipation factor η appeared at high temperatures and low strain rates; the shaded part is the instability zone. It can be seen from the hot working diagram at these three different strains that strain had a significant effect on the peak value of the dissipation rate and the instability region, and underwent regular changes. When the strain was 0.3, the instability region was mainly at low temperatures and high strain rates. When the strain was 0.5, the low-temperature and high-strain-rate region in the instability region slightly increased in size, but the high-temperature and high-strain-rate region significantly increased in size. When the strain was 0.7, the low-temperature and high-strain-rate region and the high-temperature and high-strain-rate region continued to increase in size, and the two regions merged. The instability zone also enlarged but remained in the region corresponding to a high strain rate while the strain increased. However, the peak power dissipation coefficient reduced with rising strain, from 0.56 at a strain of 0.3 to 0.49 at a strain of 0.7. The dissipation factor η represents the degree of DRX during high-temperature deformation, so the region with a higher η will have a better hot workability performance. However, a higher η also increases the potential for unstable flow.

Figure 8a,b shows the microstructure of the studied alloy under hot compression deformation at 360 °C and a strain rate of 1 s^−1^. Figure 8 shows that this region was a constant-instability zone under different strain variables, so hot processing under such conditions produced microcracks and shear bands. 

Figure 9 shows the inverse pole figure (IPF) and grain size distribution diagram under hot compression deformation at 480 °C and strain rates of 1 and 0.001 s^−1^. Different colours represent different grain orientations. In Figure 9a, grain orientation spread (GOS) was used to distinguish dynamically recrystallized grains from deformed grains. The dynamically recrystallized grains were newly generated grains without obvious internal deformation. Therefore, dynamically recrystallized grains with GOS < 2° were selected to generate Figure 9b,c. The volume fraction of DRX was 68.3%. The average grain size of DRX was 6.81 μm. Figure 9e,f shows that the DRX volume fraction was 75.8% at 480 °C and a strain rate of 0.001 s^−1^, and the average dynamically recrystallized grain size was 10.54 μm. As the strain rate decreased, DRX occurred for a longer time. By comparing Figure 9b with Figure 9e,f, it is obvious that as the DRX volume fraction increased, the grain size increased significantly. At this time, nearly complete DRX occurred in the deformed matrix [26]. DRX is a favourable deformation mechanism in hot deformation and can improve the hot working performance of an alloy, which confirms the deformation mechanism of the safe zone in the hot working diagram in Figure 7. Combined, the results of the microstructure analysis and the hot working diagram show that the optimal hot working conditions of the studied alloy were a temperature of 400–480 °C and a strain rate of 0.01–0.001 s^−1^.

## 4. Conclusions

In this experiment, the hot deformation behaviour of Mg-9Gd-4Y-2Zn-0.5Zr alloy at a temperature of 360–480 °C and strain rates of 0.001–1 s^−1^ was characterized by flow curves, constitutive equations and hot processing maps. According to the constitutive equation and hot processing maps, the following conclusions were drawn:The effect of deformation softening can be improved by increasing deformation temperature and decreasing strain rate. The increase in deformation temperature provides more energy and higher grain boundary mobility, which lead to the nucleation and growth of dynamically recrystallized grains and reduce the flow stress level. The dislocations fully spread to achieve reorganization while the strain rate decreases, thus increasing the DRX degree and softening effect. Shear bands and microcracks occurred in the Mg-9Gd-4Y-2Zn-0.5Zr alloy deformed at the temperature of 360–400 °C and strain rate of 0.01–1 s^−1^.Strain has a great influence on material constants, and the constitutive equation constructed by polynomial fitting of material constants corresponding to different strain variables is:
$$\dot{\varepsilon }=A_{\varepsilon }{\left[\sin h\left(\alpha_{\varepsilon }\sigma \right)\right]}^{n_{\varepsilon }}exp\left(-Q_{\varepsilon }/RT\right)$$
$$\alpha_{\varepsilon }=-0.04808\varepsilon^{5}+0.15574\varepsilon^{4}-0.20211\varepsilon^{3}+0.1232\varepsilon^{2}-0.02775\varepsilon +0.01222$$
$$n_{\varepsilon }=-21.15385\varepsilon^{5}+81.52681\varepsilon^{4}-113.52273\varepsilon^{3}+73.72203\varepsilon^{2}-23.49226\varepsilon +7.645$$
$$Q_{\varepsilon }=-30666.98718\varepsilon^{5}+16136.29079\varepsilon^{4}+61275.794\varepsilon^{3}-59843.06454\varepsilon^{2}\;+13822.3636\varepsilon +224105.14417$$
$$lnA_{\varepsilon }=33.78365\varepsilon^{5}-111.68539\varepsilon^{4}+138.10725\varepsilon^{3}-76.67347\varepsilon^{2}+18.49031\varepsilon +\;32.47914$$We observed an obvious effect of strain on the peak power dissipation rate coefficient and instability range. With increasing strain, the dissipation coefficient of the peak power decreased, and the instability range increased gradually. According to the analysis of the hot working diagram, the optimum hot working conditions were a temperature of 400–480 °C and a strain rate of 0.001–0.01 s−1.

## Figures and Tables

**Figure 1 materials-15-01779-f001:**
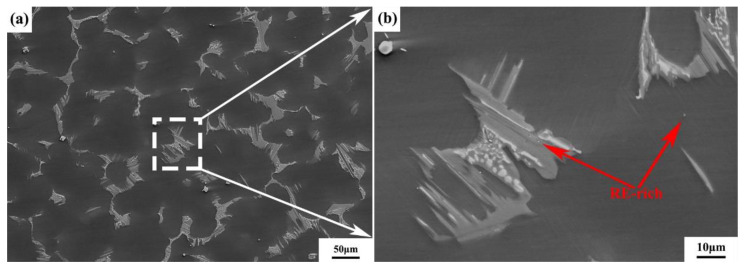
Original microstructure of Mg-9Gd-4Y-2Zn-0.5Zr alloy: (**a**) SEM image; (**b**) Partial enlargement.

**Figure 2 materials-15-01779-f002:**
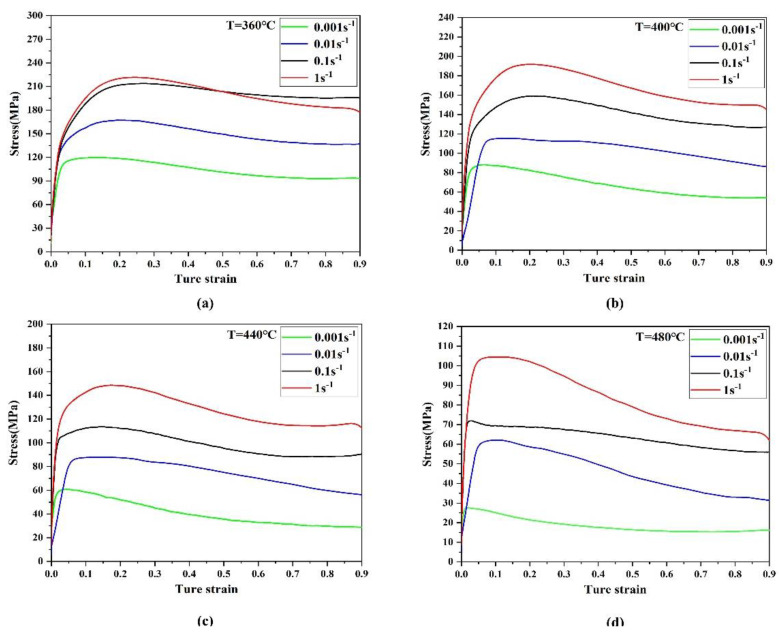
True stress-true strain curves of Mg-9Gd-4Y-2Zn-0.5Zr alloy under different deformation conditions: (**a**) 360 °C; (**b**) 400 °C; (**c**) 440 °C; (**d**) 480 °C.

**Figure 3 materials-15-01779-f003:**
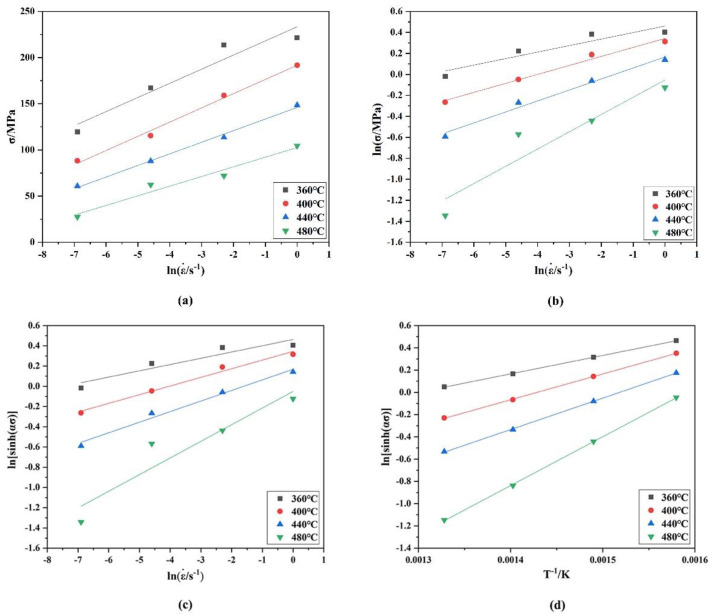
Relationship between strain rate and stress value of alloy at different deformation temperatures: (**a**)$\;\sigma -ln\dot{\varepsilon }$; (**b**) $ln\sigma -ln\dot{\varepsilon }.$ Relationship curves between (**c**)$\;ln\left[sinh\left(\alpha \sigma \right)\;\right]-ln\dot{\varepsilon }$ and (**d**) ln[sinh(ασ)]−(1/T) under different deformation parameters.

**Figure 4 materials-15-01779-f004:**
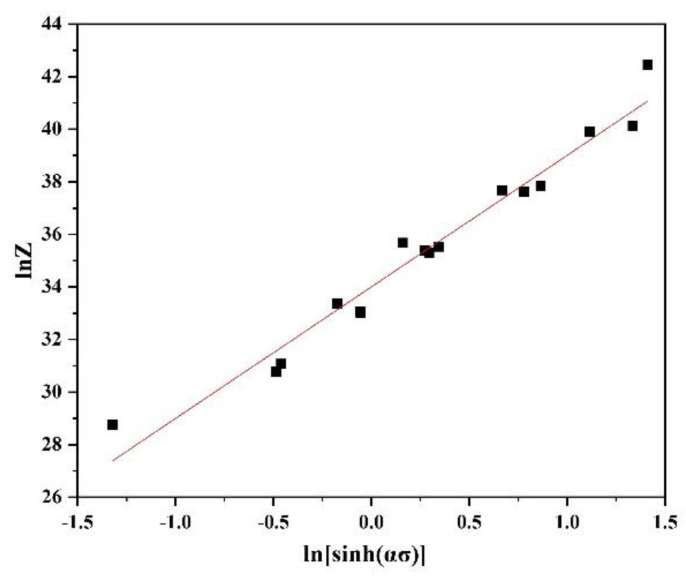
Relationship curve between *lnZ* and *ln*[*sinh*(*ασ*)].

**Figure 5 materials-15-01779-f005:**
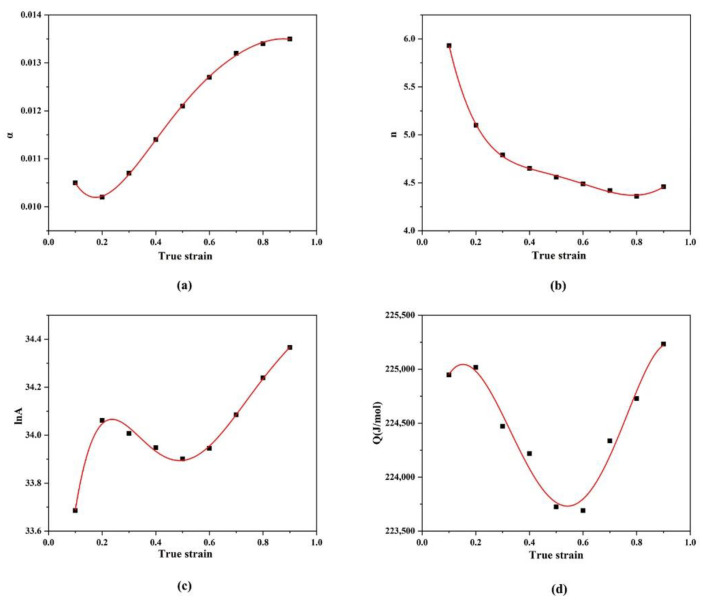
The relationship between material constants (**a**) α, (**b**) *n*, (**c**) *lnA*, (**d**) *Q* and strain under different stresses.

**Figure 6 materials-15-01779-f006:**
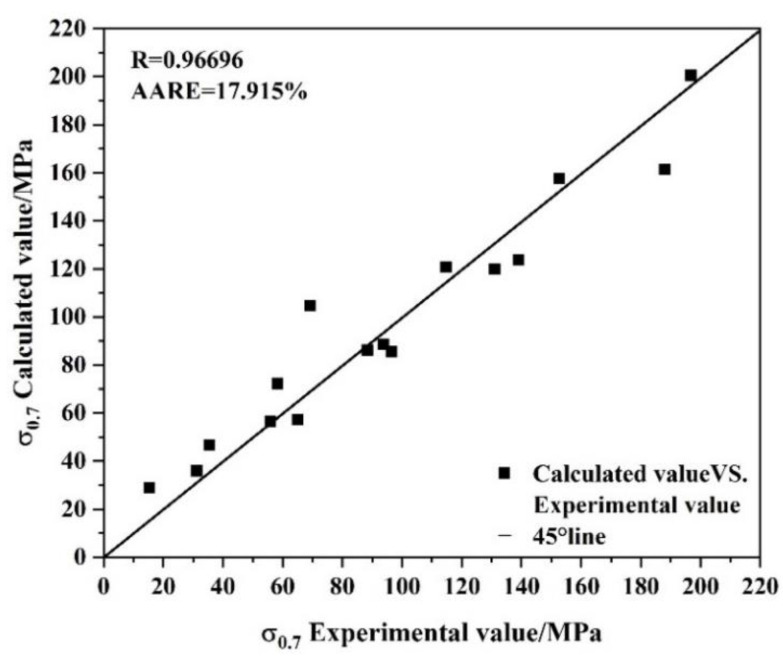
Relationship between experimental and calculated values of *σ*_0.7_.

**Figure 7 materials-15-01779-f007:**
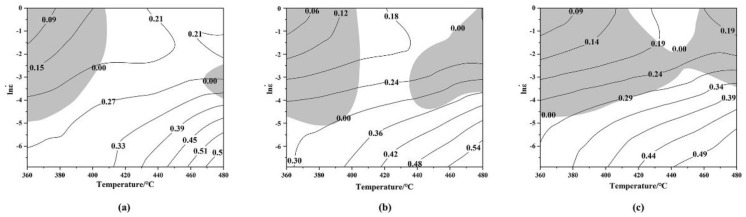
Processing maps of Mg-9Gd-4Y-2Zn-0.5Zr alloy: (**a**) *ε* = 0.3 (**b**) *ε* = 0.5; (**c**) *ε* = 0.7.

**Figure 8 materials-15-01779-f008:**
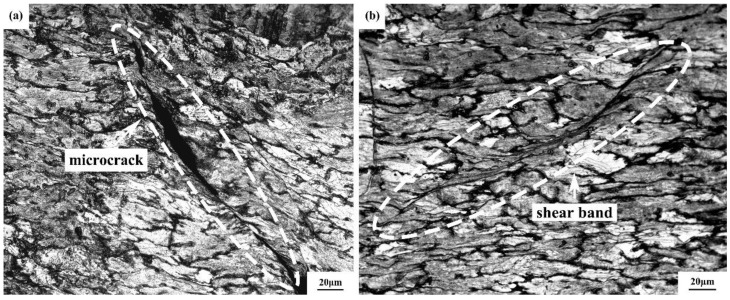
Microstructure of Mg-9Gd-4Y-2Zn-0.5Zr alloy at a deformation temperature of 360 °C and strain rate of 1 s^−1^: (**a**) microcrack; (**b**) shear band.

**Figure 9 materials-15-01779-f009:**
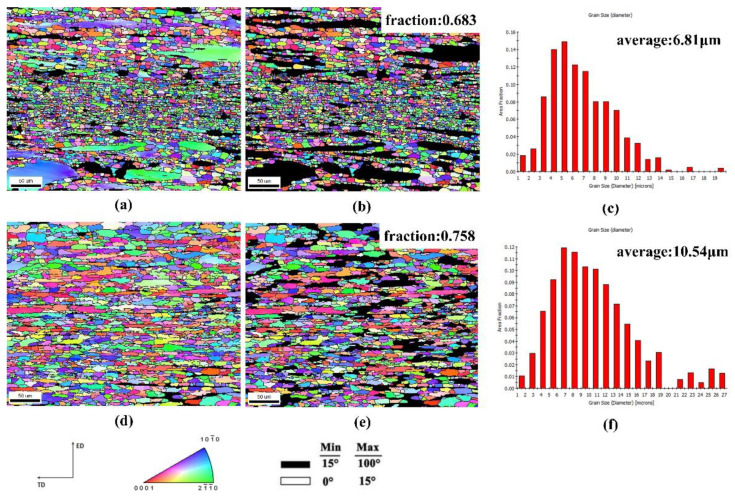
EBSD IPF maps and grain size distribution maps of samples: (**a**–**c**) 480 °C, 1 s^−1^; (**d**–**f**) 480 °C, 0.001 s^−1^. (**a**,**d**) IPF maps of full grains; (**b**,**e**) IPF maps of DRXed grains; (**c**,**f**) grain size distribution maps of samples.

**Table 1 materials-15-01779-t001:** Chemical compositions of Mg-9Gd-4Y-2Zn-0.5Zr alloy (%, mass fraction).

Element	Gd	Y	Zn	Zr	Si	Cu	Mg
wt.%	9.48	4.00	1.98	0.50	<0.01	<0.01	Bal.

## Data Availability

Data are contained within the article.

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
