# Peer review of "Hot Deformation Behaviour and Constitutive Equation of Mg-9Gd-4Y-2Zn-0.5Zr Alloy"

_materials, 2022, doi:10.3390/ma15051779_

Round 1

Reviewer 1 Report

While this work clearly shows an extensive set of results in order to provide a real and effective attempt to form an understanding of thermoplastic deformation and subsequent hot forming of Mg-9Gd04Y-2Zn-0.5Zr there is still must work required before this paper can be accepted.

You need to go through your entire manuscript and make sure you have added all the correct references in the text. For example in the sentence on line 43 you refer to Sun Z et al. but there is no reference number given in the sentence. It appears in the following sentence which is too late.

Your Experimental set up section needs significant revisions. There are major gaps present. There is no mention of the tools/instruments/techniques that you used to record images, be they optical, SEM or EBSD. All of this information needs to be added.

You present what most would consider results in your experimental set up section. It is highly recommended that you reorganize this section to remove Figure 1 and table 1 and place them in results where they belong.

The tensile curves in Figure 2 appear to show some irregularities at the start of some of the curves. For example in Figure 2a the 0.001s-1 curve, Figure 2b the  0.1 and 0.01s-1 curves and also some curves in figures c and d all have this problem. It appears that there was not sufficient pre-tension on the sample grips in your tensile testing machine and thus upon the start of a test the slack in the sample set up may have caused this issue. With this in mind, can you confidently claim the accuracy of those effected curves’ yield point and UTS? With this slack present it is very likely that those values are not accurate.

I recommend that those curves be replaced since your work is centred on empirical values in order to produce your equations.

Author Response

请参阅附件。

Reviewer 2 Report

Dear collegues,

I would like to ask you to make a conclusion about the possibility of using the calculated constants for the studied material in work for other magnesium alloys. Although you have demonstrated the calculation approach at work. Can researchers use a similar approach to develop hot deformation technology for magnesium alloys of various chemical compositions?

Best regards

Reviewer 3 Report

The article is devoted to a topical issue, namely the study of the behavior of Mg-9Gd-4Y-2Zn-0.5Zr alloy under hot plastic deformation. The study used modern research equipment Gleeble 3800. Research results can be useful both to other researchers and the industry. The article is written concisely, competently, scientifically sound. However, while reviewing it, several questions and comments arose.
1. Section 2 "Methods" does not indicate the equipment on which metallographic studies were carried out. How was the sample preparation done? How was the average grain size of 70 µm determined?
2. In Figure 1, I propose to indicate the phases. This will be useful to the reader.
3. In paragraph 3.4. the results of research and analysis of the microstructure of deformed samples are presented. However, the article does not indicate that software was used to analyse and process microstructure images. This information needs to be added.
4. Lines 219, 220, 221 and 223 contain the equations obtained by approximating the graphs shown in Fig.5. Why is a fifth-degree polynomial chosen in all equations? What are the correlation coefficients for these equations?
5. In the conclusions in lines 336-337, you need to indicate the alloy, temperature range and strain rates that were studied.

Round 2

Reviewer 1 Report

Thank you for your attempts to correct your paper.

However, you have still not provided sufficient information in your experimental procedures sections regarding you microscopy investigations. It is not sufficient that you only mention the names of the equipment that you used. You must also explain to the reader how you used those pieces of equipment so that if they wished to reproduce your experiments they are able to. So please state, what types of SEM imaging you used, the kV information also. You must also explain how you prepared your samples for EBSD. What are the settings you used for EBSD?

Reviewer 3 Report

The authors of the article have eliminated the remarks I indicated. The article can be accepted in this form.

Author Response

Thank you very much for your time. We highly appreciate the detailed valuable comments of our manuscript of ‘materials-1587456’. The suggestions are quite helpful for us.